# Let data talk: data-regularized operator learning theory for inverse problems

## Abstract

Regularization plays a critical role in incorporating prior information into inverse problems. While numerous deep learning methods have been proposed to tackle inverse problems, the strategic placement of regularization remains a crucial consideration. In this article, we introduce an innovative approach known as "data-regularized operator learning" (DaROL) method, specifically designed to address the regularization of inverse problems. In comparison to typical methods that impose regularization though the training of neural networks, the DaROL method trains a neural network on data that are regularized through well-established techniques including the Lasso regularization method and the Bayesian inference. Our DaROL method offers flexibility across various frameworks, and features a simplified structure that clearly delineates the processes of regularization and neural network training. In addition, we demonstrate that training a neural network on regularized data is equivalent to supervised learning for a regularized inverse mapping. Furthermore, we provide sufficient conditions for the smoothness of such a regularized inverse mapping and estimate the learning error with regard to neural network size and the number of training samples.

## 1 Introduction

Partial differential equations (PDEs) are fundamental modeling tools in science and engineering which play a central role in scientific computing and applied mathematics. Solving a PDE entails seeking a solution given PDE information, such as coefficient functions, boundary conditions, initial conditions, and loading sources. Conversely, solving inverse PDE problems involves reconstructing PDE information from measurements of the PDE solutions (Isakov, 1992). These two problems are commonly referred to as the forward and inverse problems respectively. In an abstract way, the forward problem can be formulated as

$$m = f(a), \tag{1}$$

where $f : \mathcal{A} \ni a \mapsto m \in \mathcal{M}$ is the forward mapping, $\mathcal{A}$ and $\mathcal{M}$ denote the spaces of the parameter of interest (PoI) and PDE solution measurement, respectively. The input $a$ represents the PoI, such as the initial conditions, the boundary conditions, or the parameter functions of a specific PDE. Typically, the output $m$ could be the PDE solution or (multiple) measurements of the PDE solution. For simplicity, we assume that a fixed discretization method is employed such that both $\mathcal{A}$ and $\mathcal{M}$ are the subspaces within Euclidean spaces $\mathbb{R}^{d_a}$ and $\mathbb{R}^{d_m}$, respectively, where $d_a$ and $d_m$ denote the ambient dimensions of the spaces $\mathcal{A}$ and $\mathcal{M}$.

In contrast to the forward problem equation 1, the inverse problem seeks to reconstruct partial knowledge of the PDE from the measurement data of the PDE solution. The inverse problem can be formulated as follows.

$$a = f^{-1}(m), \quad \text{where} \quad f^{-1} : m \mapsto a. \tag{2}$$

The inverse problem equation 2 is generally more challenging than the forward problem equation 1 due to ill-posedness of the inverse problem, meaning that even minor noise in measurements can result in considerable errors in reconstructing the parameter $a$. This issue is exacerbated when only partial measurements are accessible due to economic constraints, potentially leading to multiple plausible values of $a$ corresponding to the

same partial measurements, rendering the inverse map $f^{-1}$ ill-posed. To tackle the challenge of ill-posedness and achieve accurate solutions to inverse problems, it is imperative to employ suitable regularization techniques that integrate prior knowledge of the PoI. Two prevalent paradigms for regularization in solving inverse problems are variational regularization methods (Ito & Jin, 2014) and Bayesian inference methods (Biegler et al., 2011). Variational regularization introduces a penalty term into the loss function to encourage specific characteristics of the PDE parameters, such as sparsity and sharp edges. In contrast, Bayesian inference considers a prior probability distribution that can generate samples consistent with common understanding of the PDE parameters. Recently, the concept of applying partial regularization to the parameter space has been proposed (Wittmer & Bui-Thanh, 2021; Nguyen et al., 2022).

Deep learning has recently emerged as the leading approach for a diverse array of machine learning tasks (Graves et al., 2013; Krizhevsky et al., 2012; Brown et al., 2020), drawing considerable attention from the scientific computing community. This attention has led to the development of numerous deep learning methods tailored for solving partial differential equations (PDEs). These methods are highly regarded for their ability to handle nonlinearity and high-dimensional problems with flexibility (Han et al., 2018). Common neural network (NN) approaches include the Deep Ritz method (Yu et al., 2018), Physics-informed neural networks (PINNs) (Raissi et al., 2019), DeepONet (Lu et al., 2019), and other related techniques (Chen et al., 2023; Brandstetter et al., 2022).

Although deep learning methods have demonstrated empirical success in tackling forward PDEs, their application to solving inverse PDEs remains an evolving area. Recent years have seen the emergence of numerous methods tailored for various applications, spanning photoacoustic tomography (Hauptmann et al., 2018), X-ray tomography (Jin et al., 2017), seismic imaging (Jiang et al., 2022; Ding et al., 2022), travel-time tomography (Fan & Ying, 2023), and general inverse problems (Arridge et al., 2019; Ongie et al., 2020; Berg & Nyström, 2017; Bar & Sochen, 2019; Molinaro et al., 2023; Ong et al., 2022). Typical deep learning methods leverage a neural network, denoted as $\phi_\theta$, as a versatile surrogate to approximate the inverse map $f^{-1}$ via offline training and online prediction. During offline training, neural network weights $\theta$ are adjusted by minimizing the loss function over the training data, typically composed of input-output pairs of the target function. In the online stage, the neural network surrogate function is invoked to derive reconstructed parameters from any measurement data, effectively circumventing the need for the inverse map $f^{-1}$ and obviating the necessity for solving PDEs. While deep learning methods have achieved success in many inverse problems (Fan & Ying, 2023; Molinaro et al., 2023; Jiang et al., 2022; Pu & Chen, 2023; Haltmeier & Nguyen, 2020), it's worth noting that a naive implementation of neural networks can fail, even for the simplest inverse problems (Maass, 2019). Additionally, they may suffer from poor generalization when dealing with severely ill-posed inverse problems (Ong et al., 2022). This phenomenon aligns with an inherent feature of neural network approximation known as the frequency principle (Xu et al., 2019), meaning that neural networks tend to fit training data with low-frequency functions. Such bias can lead to excessively long training times for solving ill-posed inverse problems. Hence, there is a growing necessity to integrate deep learning methods with prior information and regularization techniques to mitigate the challenges posed by ill-posedness.

Several methodologies have emerged to merge deep learning with prior information in tackling inverse problems. In certain studies (Dittmer et al., 2020; Obmann et al., 2020; Ding et al., 2022; Heckel, 2019), regularization is accomplished by employing neural networks to parameterize the PoI, facilitating a concise representation of specific parameter characteristics such as sparsity. Another research direction (Kobler et al., 2021; Alberti et al., 2021; Chung & Español, 2017; Mukherjee et al., 2021) concentrates on variation regularization with a penalty term that is parametrized by a neural network. In the work of (Afkham et al., 2021), neural networks are used to learn the regularization parameter of the regularized variational problem. In summary, these deep learning methods for inverse problems apply regularization through the architecture of inverse problem solving, i.e. the PoI itself or the penalty term. Neural networks are adept at either parametrizing the PoI, the regularizer, or the regularization coefficient. While these methods naturally align with variational regularization for addressing inverse problems, they may be less suited for alternative frameworks like Bayesian inversion.

We propose an innovative approach that enforces regularization through the data instead of the PoI, the penalty term, or a prior distribution in the Bayesian setting. We introduce implicit regularization via training

a neural network on the regularized data to approximate the inverse map. We refer to our approach as "data-regularized operator learning" (DaROL) methods.

(1) Flexibility and Applicability: DaROL offers versatility, applicable across various frameworks. It utilizes traditional regularized inverse problem solvers to generate regularized data, including Lasso regularization and Bayesian inference methods. Once data are generated, neural networks or other deep learning models like Generative Adversarial Networks (GANs), Variational AutoEncoders (VAEs), or Denoising Diffusion Probabilistic Models can be trained. In contrast, methods employing regularization during training may encounter computational limitations, especially when differentiating neural networks to parametrize the PoI or the penalty term. (2) Simplicity and Efficiency: DaROL facilitates a straightforward and efficient online computation process. Although offline computation for data regularization may be substantial, once training data is regularized and training on regularized data is completed, solving any inverse problem becomes as simple as evaluating the trained neural network operator at the corresponding measurement data. In comparison, other existing works (Dittmer et al., 2020; Kobler et al., 2021) either requires a re-training of neural network or solving a variational problem during the online stage. (3) Theoretical Analysis: DaROL methods decouple regularization in the data generation stage from NN training in the optimization stage. In contrast, other methods intertwine NN training with regularization, posing additional challenges for theoretical analysis. In our case, we can analyze DaROL and provide theoretical estimates of the learning error concerning NN size and the number of training samples.

In this paper, we conducted a comprehensive theoretical investigation into the accuracy of the DaROL methods. Our approach began with demonstrating that the training of a neural network on data subjected to regularization leads to the acquisition of a regularized inverse map denoted as $f_{\text{reg}}^{-1}$. Subsequently, we delve into an analysis of the smoothness and regularity of this map, followed by the establishment of a learning error estimate for the neural network's approximation of $f_{\text{reg}}^{-1}$. We explore two common techniques for data regularization: the LASSO regularization and the Bayesian inference regularization. Ultimately, our paper concludes with the presentation of learning error estimates for these two distinct regularization methods.

**Main Contributions.** Our primary contributions can be succinctly summarized as follows:

- **DaROL Method:** We introduce an innovative approach known as the DaROL method, which seamlessly combines deep learning with regularization techniques to address inverse problems. In contrast to the other methods, our approach offers greater flexibility across various frameworks, accelerates the resolution of inverse problems, and simplifies the analytical process.

- **Diverse Regularization Techniques:** We explore two distinct data regularization techniques, encompassing both Lasso regularization and the Bayesian inference framework. We delve into the analysis of the smoothness of the regularized inverse map $f_{\text{reg}}^{-1}$ for both linear inverse problems (as demonstrated in Theorem 1) and nonlinear inverse problems (as evidenced by Theorem 2). Sufficient conditions are provided to show the Lipschitz property of different regularized inverse operator.

- **Learning Error Analysis:** We conduct a comprehensive analysis of the approximation error and generalization error inherent in a neural network operator surrogate for the regularized inverse map (as demonstrated in Theorem 6). Additionally, we furnish explicit estimates pertaining to the learning error of the DaROL method, taking into account factors such as neural network size and the volume of training data.

**Organization.** The remainder of this paper is structured as follows: In Section 2, we introduce the neural network architecture and DaROL method. In Section 2.3, we apply DaROL method to a linear inverse problem with $l_1$ regularization and to a nonlinear inverse problem under the Bayesian setting. In Section 3, we analyze the learning error of the DaROL method.

## 2 DaROL: data-regularized operator learning

### 2.1 Preliminaries

In this section, we will outline the standard training process for a neural network designed to approximate a general nonlinear operator $\Phi : \mathcal{X} \to \mathcal{Y}$ with Lipschitz continuity. [1] We will also establish a generalization error estimate. We aim to approximate a target operator $\Phi$ of Lipschitz continuity using a neural network function $f : \mathbb{R}^{d_x} \to \mathbb{R}$ that employs the ReLU (Rectified Linear Unit) activation function. Our work can be generalized to other classes of continuous functions, including Hölder class and Sobolev class functions.

A ReLU neural network function with $L$ layers and width $p$ is represented by the following formula:

$$f(x) = W_L \phi^{(L)} \circ \phi^{(L-1)} \circ \cdots \circ \phi^{(1)}(x) + b_L \,, \tag{3}$$

where each layer $\phi^{(i)} : \mathbb{R}^{p_{i-1}} \to \mathbb{R}^{p_i}$ consists of an affine transformation and the entrywise nonlinear ReLU activation function $\phi^{(i)}(x) = \sigma(W_i x + b_i) \,, i = 1, \ldots, L \,.$ Here $\sigma(x) = \max\{x, 0\}$ represents the ReLU activation function, and $W_i \in \mathbb{R}^{p_i \times p_{i-1}}$ and $b_i \in \mathbb{R}^{p_i}$ are the weight matrices and bias vectors for each layer. The ReLU function is applied pointwise to all entries of the input vector in each layer. Assuming, without loss of generality, that $p_i = p$ for $i = 0, \cdots, L-1$ and $p_L = 1$ to match the output dimension of the neural network function $f$. In cases where the widths of each layer differ, we can always select $p$ as the largest width and introduce zero paddings in the weight matrices and bias vectors.

We define a class of neural network functions with $L$ layers and width $p$ that have uniformly bounded output:

**Definition 1** (ReLU Neural Network Function Class)**.**

$$\mathcal{F}_{NN}^0(L, p, M) = \{f : \mathbb{R}^{p_0} \to \mathbb{R} \mid \qquad f \text{ is in the form of equation 3,}$$
$$\text{and } \max_x \|f(x)\| \le M \,, \ \forall x \in \mathbb{R}^{p_0}\}. \tag{4}$$

To approximate a nonlinear operator with $d_a$ outputs, we stack the neural network functions described in equation 4 and define a class of neural network operators with $L$ layers and width $pd_a$ that have bounded output as follows:

**Definition 2** (ReLU Neural Network Operator Class)**.**

$$\mathcal{F}_{NN}(d_a, L, pd_a, M) = \{f : \mathbb{R}^{p_0} \to \mathbb{R}^{d_a} \mid \qquad f(x) = [f_1(x), \ldots, f_{d_a}(x)]^\top \,,$$
$$f_i \in \mathcal{F}_{NN}^0(L, p, M) \,, \ \forall i = 1, \ldots, d_a\}. \tag{5}$$

It's worth noting that the width of $\mathcal{F}_{\text{NN}}$ is always a multiple of its output dimension $d_a$ since we stack subnetworks $f_i$ with the same width. We may use the notation $\mathcal{F}_{\text{NN}}^0$ and $\mathcal{F}_{\text{NN}}$ without specifying their parameters when no ambiguity arises.

We engage in supervised learning with the goal of approximating a target operator $\Phi : \mathcal{X} \to \mathcal{Y}$ using a neural network operator $\phi_\theta$ from the class $\mathcal{F}_{\text{NN}}$. Here, $\theta$ represents all the trainable parameters, including the weight matrices and bias vectors in all subnetworks $\phi_i$. In typical supervised learning scenarios, we assume access to a training dataset comprising $n$ (noisy) samples of the target operator $\Phi$, given by:

$$\mathcal{S}_{\text{train}} = \{z_i := (x_i, y_i) \mid y_i = \Phi(x_i) + \varepsilon_i \,, i = 1, \ldots, n\} \,. \tag{6}$$

Here $x_i$ are i.i.d (independent and identically distributed) samples of a random distribution over $\Phi$, and $\varepsilon_i$ denotes i.i.d. noise. The training of the neural network $\phi_\theta$ involves minimizing a specific loss function over the training data $\mathcal{S}_{\text{train}}$:

$$\theta^* \in \arg\min_\theta \frac{1}{n} \sum_{i=1}^n l(z_i, \phi_\theta) \,. \tag{7}$$

In this equation, $l(z, \phi_\theta)$ denotes the loss function associated with $\phi_\theta$ applied to a single data pair $z = (x, y)$. Common examples of loss functions include the square loss, defined as $l(z, \phi_\theta) = \|y - \phi_\theta(x)\|_2^2$. Upon completion of the training process, the trained model $\phi_{\theta^*}$ can be used as a surrogate for the target operator $\Phi$.

---

[1]See Definition 3 in the Appendix.

## 2.2 Regularization of the training data

Deep operator learning has proven to be highly effective in tackling the forward problem. This success can be attributed to the generally favorable regularity properties displayed by the forward PDE, leading to a well-behaved forward map, denoted as $f$. Moreover, obtaining training data for the forward problem is straightforward. It simply involves evaluating the forward map at randomly generated points, represented as $x_i$. This process is equivalent to solving the corresponding forward PDE once, making data collection a relatively uncomplicated task.

Transitioning from forward operator learning to inverse operator learning poses unique challenges. In most cases, inverse problems are ill-posed, exhibiting lower regularity compared to their forward counterparts. Additionally, the inverse operator, denoted as $f^{-1}$, may not be well-defined when only partial measurements are available. Nevertheless, deep learning has been employed for solving inverse problems through a supervised learning approach, similar to the one used for the forward problem. The data generation process for the inverse problem training is typically identical to that of the forward operator $f$ for practicality and cost-effectiveness. This process involves generating a random function, denoted as $a$, and evaluating the forward map $f(a)$, which is equivalent to solving a PDE once. Subsequently, a training data pair is created by introducing noise to the measurement, resulting in the following training dataset:

$$\mathcal{S}_{\text{imp}} = \{z_i := (m_i, a_i) \mid m_i = f(a_i) + \varepsilon_i, i = 1, \dots, n\}. \tag{8}$$

This dataset is referred to as "implicit" training data because the output $a_i$ of the target map $f^{-1}$ is not directly obtained by evaluating $f^{-1}$; instead, it is independently generated from a random measure that depends on both the distribution of $a_i$ and the noise. In contrast, the general supervised training dataset described in equation 6 suggests that the natural approach for learning the inverse map $f^{-1}$ should consider the "explicit" training data as follows.

$$\mathcal{S}_{\text{exp}} = \{z_i := (m_i, a_i) \mid a_i = f^{-1}(m_i), i = 1, \dots, n\}, \tag{9}$$

where $m_i$'s are i.i.d. random variables generated from a certain probability distribution, which may depend on the noise distribution. In this context, we assume that there are no noises in $a_i$ because, in practice, the noises are present only in the measurements $m_i$. Although both training datasets, equation 8 and equation 9, contain data pairs comprising measurements $m_i$ and their corresponding parameters $a_i$, the implicit setup equation 8 is prevalent in the majority of simulations involving deep learning for inverse operators. This is primarily because generating a single data pair in equation 9 requires direct evaluation of $f^{-1}$, which is not readily available in practice. It has been shown that training a neural network on the latter case equation 9 has a data complexity that depends on the Lipschitz estimate of $f^{-1}$ (Chen et al., 2022), but for most inverse problem the Lipschitz estimate of $f^{-1}$ is unobtainable or unbounded (Alessandrini, 1988) due to the ill-posedness nature of inverse problems. The question thus remains open whether a neural network can effectively learn $f^{-1}$ from these training datasets.

To address the ill-posedness, incorporating prior information about the PDE parameter $a$ is crucial. This information is typically integrated into the neural network training process. There are two main approaches for doing this: (1) Deep Image Prior: in this method, the image $a(x)$ is parametrized as a neural network and thus implicit regularization via such an architecture is imposed to solve the inverse problems (Dittmer et al., 2020; Berg & Nyström, 2021; Haltmeier & Nguyen, 2023). (2) Variational Regularization: Another approach involves adding a regularization term $\mathcal{R}(a)$ to the training loss, which characterizes the prior information. (Ito & Jin, 2014; Nguyen & Bui-Thanh, 2021). This regularization term guides the learning process and enforces constraints that align with the prior knowledge about the parameter $a$. Both of these approaches have shown effectiveness in producing reasonable reconstructions. However, there is a lack of theoretical understanding about these methods, which limits further improvements in terms of accuracy and performance.

We propose an alternative approach that combines explicit training with regularization. Specifically, we consider the following regularized training dataset:

$$\mathcal{S}_{\text{reg}} = \{z_i := (m_i, \hat{a}_i) \mid \hat{a}_i = f_{\text{reg}}^{-1}(m_i), i = 1, \dots, n\}, \tag{10}$$

where $m_i = f(a_i) + \varepsilon_i$ are measurement data, $a_i$'s are i.i.d. samples of the PoI, and $\hat{a}_i$'s are the regularized PoI. In essence, the regularized dataset $\mathcal{S}_{\text{reg}}$ has i.i.d. generated input, and the outputs are obtained through the

evaluation of a "regularized inverse operator," denoted as $f_{\text{reg}}^{-1}$. The advantages of training on the regularized dataset over other strategies have been summarized in Table 1. While combining regularization with inverse problem solving is the key to tackle the ill-posedness, theoretical understanding is unknown for training on explicit data with regularization via architecture or penalty term. Our work considers an alternative regularization route via the regularized training data, and provides theoretical understanding of such strategy.

Table 1: Comparison of various of features between different neural network training strategies

| Type of training data | Explicit data equation 9 | Explicit data with regularization | Regularized data equation 10 |
|---|---|---|---|
| regularity of the target map | No, (Alessandrini, 1988) | No | Yes, this work. |
| additional computation during training | No | Yes | No |
| theoretical understanding | Yes, (Györfi et al., 2002) | No | Yes, this work |

The regularized data can be computed using standard regularization techniques, such as variational regularization (Ito & Jin, 2014) or Bayesian approaches (Calvetti & Somersalo, 2018). Through the regularization of the training data, the original ill-posed target map $f^{-1}$ is replaced by a well-posed one $f_{\text{reg}}^{-1}$. Compared to the explicit training dataset in *equation* 9, training on a regularized dataset is presumably easier because the regularized inverse $f_{\text{reg}}^{-1}$ has smaller modulus of continuity for linear inverse problems (Hofmann et al., 2008). As a result, it is easier for a neural network to learn the regularized inverse operator, making this approach advantageous for ill-posed inverse problems.

## 2.3 Proposed method

In this section, we explore two common methods for generating regularized data: the variational regularization method and the Bayesian inference approach. In general, the Bayesian inference approach is more flexible comparing to the variational method, because an explicit penalty term that encodes the prior information is needed for the variational method, while recent generative models (Ho et al., 2020) can encode prior information automatically. Specifically, the Bayesian inference approach is equivalent to LASSO problem if the prior distribution is chosen to be independent double exponential distribution (Park & Casella, 2008). To illustrate these methods, we will use a linear inverse problem, specifically LASSO, for variational regularization, and a nonlinear inverse problem for the Bayesian inference approach. These examples will help demonstrate the practical application of these regularization techniques in different contexts.

### 2.3.1 Lasso regularized data

We define the regularized inverse as follows, considering two compact metric spaces, $\mathcal{M}$ and $\mathcal{A}$:

$$f_{\text{reg}}^{-1} : \mathcal{M} \ni m \mapsto \hat{a} \in \mathcal{A}, \tag{11}$$

where $\hat{a}$ represents the unique solution to the following optimization problem:

$$\hat{a} = \arg\min_{a \in \mathcal{A}} \|f(a) - m\|^2 + \lambda \mathcal{R}(a). \tag{12}$$

To ensure the existence and uniqueness of $\hat{a}$, we assume $\mathcal{A}$ is a convex domain and $\mathcal{R}$ is a convex function. In addition to these assumptions, further conditions may be required for $f$ to guarantee the uniqueness of $\hat{a}$.

In the upcoming discussion, we focus on a linear forward map, which can be expressed as $f(a) = Aa$, and a regularization term similar to LASSO. For more extensive information on different functional forms of $f$ and alternative regularization terms, such as the $l^2$ norm and the Frobenius norm, we are referred to (Vaiter et al., 2015; 2017).

The LASSO regularization problem is formulated as follows:

$$\hat{x} = \arg\min_{x \in \mathcal{X}} \frac{1}{2}\|Ax - b\|^2 + \lambda\|x\|_1 \, . \tag{13}$$

Here $\lambda > 0$ is a tuning parameter. The LASSO method was initially proposed in (Santosa & Symes, 1986) and introduced in (Tibshirani, 1996), primarily for applications in signal processing. This method played a pivotal role in the development of the field of *compressive sensing* as described in (Donoho, 2006). Its properties, including uniqueness and sensitivity concerning the tuning parameter, have been extensively studied, as evidenced in (Candes & Tao, 2005; Candes et al., 2006). Recent research has also explored the stability of the LASSO problem in relation to the data vector $b$, as detailed in (Berk et al., 2022). A similar analysis can be applied to the square-root LASSO problem, as discussed in (Berk et al., 2023).

In the context of operator learning for inverse problems, the focus often shifts towards understanding the regularity of the mapping $b \mapsto \hat{x}$. It has been demonstrated that this mapping exhibits certain regularity properties. The following condition helps ensure both uniqueness and stability in this context.

**Assumption 1** (Assumption 4.4 in (Berk et al., 2022) (non-degeneracy condition)). *For a solution $\hat{x}$ of equation 13, and $I = supp(\hat{x})$, we have (i) $A_I$ has full column rank $|I|$; (ii) $\|A_{I^C}^\top(b - A_I\hat{x}_I)\|_\infty < \lambda$.*

Here $A_I$ represents the submatrix of $A$ containing columns of $A$ within the index set $I$, and $I^C$ denotes the complement column indices of $I$. Condition (i) in the non-degeneracy condition is necessary for the unique reconstruction of $\hat{a}$. In condition (ii), $\bar{r} := (b - A_I\hat{x}_I)$ models the noises in the measurement data $b$. Notice that

$$\|A_{I^C}^\top(b - A_I\hat{x}_I)\|_\infty = \max_{j \in I^C}|\langle a_j, r\rangle| \leq \|r\|_2 \max_{j \in I^C}\|a_j\| \leq \|r\|_2\|A\|_F \, ,$$

Therefore condition (ii) holds if $\lambda > \|r\|_2\|A\|_F$. In practice, this is valid condition when $\lambda$ is chosen large enough or the noise is sufficiently small.

We will establish a global stability result for the regularized inverse map $b \mapsto \hat{x}$. Our proof draws on variational analysis as presented in (Berk et al., 2022), and it involves an SVD (Singular Value Decomposition) analysis of the submatrix $A_I$.

**Theorem 1.** *Consider the regularized inverse $f_{reg}^{-1}$ defined in equation 12. Assume that (i) $f(a) = Aa$ with $A \in \mathbb{R}^{d_m \times d_a}$ and $\mathcal{R}(\cdot) = \|\cdot\|_1$; (ii) The non-degeneracy Assumption 1 holds for all $\hat{a}$. Then the regularized map $b \mapsto \hat{x}$ is globally Lipschitz for all $b$ with the following Lipschitz constant:*

$$L \leq \frac{Cond(A)}{\sigma_{min}(A)} + \frac{1}{\sigma_{min}^2(A)} \, .$$

**Remark 1.** *When the measurement noise $\bar{r} = (b - A_I\hat{x}_I)$ is small and the regularization parameter $\lambda$ is large, the non-degeneracy condition (ii) holds. This implies that, with sufficiently small noise, the regularized inverse operator is globally Lipschitz. In other words, the regularized inverse map exhibits a stable and well-behaved behavior for a wide range of input data, making it a robust tool for solving inverse problems.*

### 2.3.2 Bayesian regularized data

The Bayesian inference method characterizes prior information about the parameter $a$ using a prior distribution denoted as $\mu_0$. The Bayesian law provides a posterior distribution $\mu^m$ of the reconstructed parameter $a$ given measurement data $m$, which can defined as follows:

$$\frac{d\mu^m(a)}{d\mu_0} = \frac{1}{Z(m)}\exp(-\Phi(a;m)) \, .$$

Here $Z(m) > 0$ is a probability normalizing constant, $\exp(-\Phi(a;m))$ is usually referred to as the likelihood function, and $-\Phi(a;m$ is called the log likelihood function.

**Remark 2.** *For a nonlinear inverse problem $m = f(a) + \varepsilon$ where $f : \mathbb{R}^{d_a} \mapsto \mathbb{R}^{d_m}$ is the forward map and the noise $\varepsilon \sim \mathcal{N}(0, \Sigma)$ follows a normal distribution with covariance matrix $\Sigma \in \mathbb{R}^{d_m \times d_m}$, the log likelihood function is $-\Phi(a;m) = -\frac{1}{2}(m - f(a))^\top \Sigma (m - f(a))$.*

To obtain posterior samples, a Markov Chain Monte Carlo (MCMC) method is often used to generate a sequence of samples that converges to a sample that follows the posterior distribution. While the Bayesian inference method provides an explicit posterior distribution and the corresponding MCMC method can generate posterior samples, it is up to the user to construct a representative solution from this distribution. The two most common choices are the Maximum a Posteriori (MAP) solution and the Conditional Mean (CM) solution, defined as follows:

$$a_{\mathrm{MAP}} = \arg\max_a \mu^m(a), \quad a_{\mathrm{CM}} = \mathbb{E}_{\mu^m}[a]$$

The MAP solution represents the most likely parameter given the data and prior information, while the CM solution represents the expected value of the parameter conditioned on the data and prior. For the simplicity of analysis and data generation, we select the CM as the regularized inverse. We define that,

$$\hat{a} := \mathbb{E}_{\mu^m}[a] = \frac{1}{Z(m)} \int_{\mathcal{A}} a \exp(-\Phi(a; m)) d\mu_0 \,. \tag{14}$$

In practical implementations, the MCMC method first generate a sequence of samples $a_1, a_2, \ldots, a_N$, and one can compute the approximated value of CM estimate $\hat{a} = \mathbb{E}_{\mu^m}[a] \approx \frac{2}{N} \sum_{i=\lfloor N/2 \rfloor+1}^{N} a_i$, by discarding the first half of the sequence and averaging the values in the second half. For details on the implementation of conditional mean estimation, please refer to the book (Kaipio & Somersalo, 2006). This approach provides an estimate of the regularized inverse based on Bayesian inference and is commonly used in practical applications. The local Lipschitz property in Hellinger distance of posterior distributions is studied in (Cotter et al., 2009). We follow a similar approach and establish in the theorem below that the regularized inverse $f_{\mathrm{reg}}^{-1}$ defined by the conditional mean equation 14 is global Lipschitz.

**Theorem 2.** *Consider the forward map $f : a \mapsto m$ and regularized inverse $f_{reg}^{-1} : m \mapsto \hat{a}$ with $\hat{a}$ as the conditional mean defined in equation 14. Suppose the following assumptions hold:*

*(i) Both the parameter space $\mathcal{A}$ and measurement space $\mathcal{M}$ are bounded.*

*(ii) $\Phi : \mathcal{A} \times \mathcal{M} \to \mathbb{R}$ is continuous.*

*(iii) There exists two functions $M_i : \mathbb{R}^+ \to \mathbb{R}, i = 1, 2$ such that for all $a \in \mathcal{A}, m, m' \in \mathcal{M}$, we have*

$$-\Phi(a; m) \leq M_1(\|a\|), \quad and \quad |\Phi(a; m) - \Phi(a; m')| \leq M_2(\|a\|)\|m - m'\| \,. \tag{15}$$

*(iv) $M_1$ and $M_2$ satisfies the following integrable conditions*

$$\exp(M_1(\| \cdot \|)) \in L^1_{\mu_0}(\mathcal{A}), \quad and \quad \exp(M_1(\| \cdot \|))M_2(\| \cdot \|) \in L^1_{\mu_0}(\mathcal{A}) \,. \tag{16}$$

*Then the regularized inverse $f_{reg}^{-1} : m \mapsto \hat{a}$ is globally Lipschitz, i.e. for all $m, m' \in \mathcal{M}$,*

$$\|f_{reg}^{-1}(m) - f_{reg}^{-1}(m')\| \leq 2e^{-2R} \sup_{a \in \mathcal{A}} \|a\| C(M_1, M_2) \|m - m'\| \,,$$

*where $R = \sup_{(a,m) \in \mathcal{A} \times \mathcal{M}} \Phi(a; m)$ and $C(M_1, M_2)$ only depends on the $L^1_{\mu_0}$ norm of $\exp(M_1(\| \cdot \|))$ and $\exp(M_1(\| \cdot \|))M_2(\| \cdot \|)$.*

**Remark 3.** *Condition (i) and (ii) are generally true in practice. Condition (iii) assumes an upper bound of the log likelihood function and a Lipschitz condition with respect to the measurement parameter $m$. Condition (iv) assumes that the likelihood function is integrable with respect to parameter $a$ and so is its derivatives with respect to $m$. Such condition ensures that the normalizing constant $Z(m)$ in equation 14 is bounded and also helps us to bound the variation of $\hat{a}$ with respect to $m$. In particular, when the forward map $f$ is polynomially bounded and Lipschitz, one can show that condition (iii) holds for polynomial functions $M_1$ and $M_2$. This further implies condition (iv) holds when the prior distribution $\mu_0$ is uniformly bounded. Similar conditions have been considered in (Cotter et al., 2009) and applied to inverse problems in fluid dynamics.*

# 3 Learning error analysis

In both cases, where the regularized data is generated using either the regularization method discussed in Section 2.3.1 or the Bayesian method described in Section 2.3.2, a neural network denoted as $\phi_\theta$ is trained on the regularized data set $\mathcal{S}_{\text{reg}}$. The training is typically carried out through the optimization process detailed in equation 7, resulting in a trained neural network, which we denote as $\phi_{\theta^*}$. This trained neural network is designed to learn and approximate the inverse operator based on the regularized data. Given measurement data $m$, the trained neural network can generate a point estimate $\phi_{\theta^*}(m)$ for the target parameter function $\hat{a} = f_{\text{reg}}^{-1}(m)$ and thus solving the inverse problems automatically with regularization.

The learning error of the trained neural network $\phi_{\theta^*}$ is a measure of the discrepancy between the true value $f_{\text{reg}}^{-1}(m)$ and the neural network's prediction $\phi_{\theta^*}(m)$ for an independent data $m$. This independent data point is typically an independent copy of the $m_i$ in the $\mathcal{S}_{\text{reg}}$ data set. The learning error is defined as:

$$\mathcal{E}_{\text{learning}} := \mathbb{E}\left[\|f_{\text{reg}}^{-1}(m) - \phi_{\theta^*}(m)\|^2\right] \leq \mathcal{E}_{\text{approximation}} + \mathcal{E}_{\text{generalization}}. \tag{17}$$

Here, we decompose the learning error into two components: approximation error and generalization error, which are defined as follows:

$$\mathcal{E}_{\text{approximation}} := \mathbb{E}_{\mathcal{S}_{\text{reg}}} \frac{1}{n} \sum_{i=1}^{n} \|f_{\text{reg}}^{-1}(m_i) - \phi_{\theta^*}(m_i)\|^2,$$

$$\mathcal{E}_{\text{generalization}} := \mathbb{E}_{\mathcal{S}_{\text{reg}}} \mathbb{E}_m \left[ \|f_{\text{reg}}^{-1}(m) - \phi_{\theta^*}(m)\|^2 - \frac{1}{n} \sum_{i=1}^{n} \|f_{\text{reg}}^{-1}(m_i) - \phi_{\theta^*}(m_i)\|^2 \right].$$

The approximation error measures the Mean Square Error (MSE) between the target function $f_{\text{reg}}^{-1}$ and the function class $\mathcal{F}_{\text{NN}}(p, L, M)$ represented by the trained neural network on the training data set. This component characterizes the approximation power of the hypothesis space of neural network operators, and it is primarily controlled by factors such as the neural network width ($p$), depth ($L$), and the choice of activation functions. The generalization error, on the other hand, is mainly controlled by the number of training samples ($n$). It represents the ability of the trained neural network to generalize its learned knowledge to independent data points. Generalization error reflects how well the neural network can make accurate predictions on data it hasn't seen during training.

## 3.1 Approximation error

For simplicity, we can consider a neural network class based on the results presented in (Shen et al., 2019) that consists of neural network functions with $L$ layers, length $p$, and activated by the ReLU function. The key insight from (Shen et al., 2019) is that such a neural network class has the capacity to approximate any continuous functions. This result provides a foundation for understanding the approximation power of neural networks in the context of our analysis.

**Theorem 3** (Theorem 1.1 of (Shen et al., 2019))**.** *Given any $p, L \in \mathbb{N}^+$, and an arbitrary continuous function $f$ defined on $[0,1]^d$, there exists a function $\phi$ implemented by a ReLU network with width $12\max\{d\lfloor p^{1/d}\rfloor, p+1\}$ and depth $12L + 14 + 2d$ such that*

$$\sup_{x \in [0,1]^d} |f(x) - \phi(x)| \leq 19\sqrt{d}w_f(p^{-2/d}L^{-2/d}),$$

*where $w_f(\cdot)$ represents the modulus of continuity function of $f$.*

Applying Theorem 3 to the regularized inverse $f_{\text{reg}}^{-1} : \mathcal{M} \to \mathcal{A}$, we can conclude the following:

**Theorem 4.** *Consider a $L_f$-Lipschitz map $f_{reg}^{-1} : \mathcal{M} \to \mathcal{A}$, where $\mathcal{M}$ is a compact subset of $\mathbb{R}^{d_m}$ and $\mathcal{A}$ is a compact subset of $\mathbb{R}^{d_a}$. There exists a ReLU neural network $\phi : \mathcal{M} \to \mathbb{R}^{d_a}$ from the neural network class $\mathcal{F}_{NN}$, defined in equation 5, with width $12d_a\max\{d_m\lfloor p^{1/d_m}\rfloor, p+1\}$ and depth $12L + 14 + 2d_m$, such that*

$$\sup_{m \in \mathcal{M}} \|f_{reg}^{-1}(m) - \phi(m)\|_2 \leq 19\sqrt{d_m d_a}L_f r_{\mathcal{M}} p^{-2/d_m} L^{-2/d_m},$$

*where $r_{\mathcal{M}}$ is the radius of $\mathcal{M}$.*

This theorem provides a concrete bound on the approximation error in terms of the key factors involved. The approximation error $\mathcal{E}_{\text{approximation}}$ is limited by the neural network size (width and depth), input and output dimensions, the Lipschitz constant $L_f$, and the radius of the input space $\mathcal{M}$. The bound is given by:

$$\mathcal{E}_{\text{approximation}} = \mathbb{E}_{\mathcal{S}_{\text{reg}}} \frac{1}{n} \sum_{i=1}^{n} \|f_{\text{reg}}^{-1}(m_i) - \phi_{\theta^*}(m_i)\|^2 \leq 361 d_m d_a L_f^2 r_{\mathcal{M}}^2 p^{-4/d_m} L^{-4/d_m} .$$

This expression quantifies the relationship between these factors and the accuracy of the approximation when using a neural network to represent the regularized inverse operator.

### 3.2 Generalization error

Denote that $\hat{a} = f_{\text{reg}}^{-1}(m)$ and $\hat{a}_i = f_{\text{reg}}^{-1}(m_i)$. We define the square loss of a function $f$ over a data pair $z = (m, \hat{a})$ as $l(z, f) = \|\hat{a} - f(m)\|^2$. Additionally, we introduce the testing loss $\mathcal{R}(\phi_{\theta^*})$ and the empirical loss as follows:

$$\mathcal{R}(\phi_{\theta^*}) := \mathbb{E}_m l(z, \phi_{\theta^*}) , \quad \hat{\mathcal{R}}(\phi_{\theta^*}) := \frac{1}{n} \sum_{i=1}^{n} l(z_i, \phi_{\theta^*}).$$

With these definitions, we can express the generalization error as:

$$\mathcal{E}_{\text{generalization}} := \mathbb{E}_{\mathcal{S}_{\text{reg}}} \mathbb{E}_m \left[ l(z, \phi_{\theta^*}) - \frac{1}{n} \sum_{i=1}^{n} l(z_i, \phi_{\theta^*}) \right] = \mathbb{E}_{\mathcal{S}_{\text{reg}}} \left[ \mathcal{R}(\phi_{\theta^*}) - \hat{\mathcal{R}}(\phi_{\theta^*}) \right] .$$

**Theorem 5.** *Consider the training dataset $\mathcal{S}_{reg}$ as given in equation 10. We make the following assumptions:*

   *(i) Each data point $m_i$ is independently generated by a random measure $p_m$ over the input space $\mathcal{M}$.*

   *(ii) The neural network class $\mathcal{F}_{NN}$ is composed of vector-valued functions $\phi$ with mappings from $\mathcal{M}$ to $\mathbb{R}^{d_a}$. Specifically, $\mathcal{F}_{NN}$ can be expressed as*

$$\mathcal{F}_{NN} = \{\phi : \mathcal{M} \to \mathbb{R}^{d_a} \mid \phi = (\phi_1, \ldots, \phi_{d_a}), \quad \phi_i \in \mathcal{F}_{NN}^0\}$$

   *(iii) The sub-network $\mathcal{F}_{NN}^0$ consists of neural network functions with a width of $p$, a depth of $L$, and weight matrices that have a uniform bounded Frobenius norm of $M_F$. Additionally, the output of these networks is bounded by $r_{\mathcal{F}_{NN}^0}$.*

*The generalization error $\mathcal{E}_{generalization}$ can be bounded as follows:*

$$\mathbb{E}_{\mathcal{S}_{reg}} \left[ \mathcal{R}(\phi_{\theta^*}) - \hat{\mathcal{R}}(\phi_{\theta^*}) \right] \leq 8\sqrt{2\ln 2} d_a (r_{\mathcal{A}} + r_{\mathcal{F}_{NN}^0}) \frac{r_{\mathcal{M}} \sqrt{L} M_F^L}{\sqrt{n}} . \tag{18}$$

Considering the same neural network class as defined in Theorem 4, each subnetwork should have a width of $12 \max\{d_m \lfloor p^{1/d_m} \rfloor, p+1\}$ and a depth of $12L + 14 + 2d_m$. As a consequence of Theorem 5, the generalization error can be bounded as follows:

$$\mathcal{E}_{\text{generalization}} \leq c_1 d_a (r_{\mathcal{A}} + r_{\mathcal{F}_{\text{NN}}^0}) \frac{r_{\mathcal{M}} \sqrt{\max\{L, d_m\}} M_F^{c_2 \max\{L, d_m\}}}{\sqrt{n}} ,$$

where $c_1, c_2 > 0$ are absolute constants.

Finally, we summarize the learning error for a Lipschitz operator in the following theorem.

**Theorem 6.** *Consider a $L_f$-Lipschitz map $f_{reg}^{-1} : \mathcal{M} \to \mathcal{A}$. If the following assumptions hold:*

   *(i) $\mathcal{M}$ is a compact subspace of $\mathbb{R}^{d_m}$;*

*(ii) the neural network class $\mathcal{F}_{NN}$ in equation 5 has weight matrices uniformly bounded by $M_F$ in Frobenius norm.*

*Then we can obtain a neural network operator $\phi_{\theta^*}$ with the training data set equation 10, from the neural network operator class $\mathcal{F}_{NN}(d_a, L, pd_a, M)$ with the following parameter choice:*

$$L = \Omega(d_m \tilde{L}), \quad p = \Omega(d_m^{\frac{d_m}{d_m-1}} \tilde{p}), \quad and \quad M = L_f r_{\mathcal{M}} + 1,$$

*where $\tilde{L}, \tilde{p} > 0$ are arbitrary constants, such that*

$$\mathcal{E}_{learning} \le c d_a d_m^{1/2} r_{\mathcal{M}}^2 \left( L_f^2 d_m^{\frac{d_m^2 - 9d_m + 4}{2d_m(d_m-1)}} \left(\tilde{p}\tilde{L}\right)^{-4/d_m} + L_f \frac{\sqrt{\tilde{L}} M_F^{\tilde{c} d_m}}{\sqrt{n}} \right).$$

*Here $c, \tilde{c} > 0$ are absolute constants. The notation $y = \Omega(x)$ means there exist $c_1, c_2 > 0$ such that $c_1 x \le y \le c_2 x$.*

**Remark 4.** *The hyperparameters $\tilde{p}$ and $\tilde{L}$ characterize the neural network size. We can observe that the approximation error decreases polynomially with increasing neural network size, represented by $\tilde{p}\tilde{L}$. In contrast, the generalization error decreases at a square root rate as the number of training samples grows. Moreover, a larger neural network depth tends to lead to poorer generalization. The Lipschitz constant $L_f$ has a quadratic impact on the approximation error and a linear effect on the generalization error.*

**Remark 5.** *Theorem 6 provides the learning error for general Lipschitz operators. However for most inverse problems that inverse operator is not Lipschitz and training on a raw data set would lead to large learning error. When combining Theorem 6 with our main results Theorem 1 and Theorem 2, one can obtain a learning error estimate for the trained neural network via the DAROL method.*

## 4 Limitations

This paper presents theoretical foundation of the data-regularized operator learning for inverse problems. However, many inverse problems are severely ill-posed in the sense that the inverse operator is not Lipschitz but only continuous with bad modulus of continuity. A famous example is the Calderón problem (Uhlmann, 2012). To tackle this issue, the approximation results for neural network in (Shen et al., 2019) or (Yarotsky, 2018) would fail and a more general approximation result is needed, which will be left for future works.

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

# A Appendix

## A.1 Preliminaries

**Definition 3.** *(Lipschitz Continuous Operator) An operator $\Phi : \mathcal{X} \to \mathcal{Y}$ is deemed Lipschitz continuous if there exists a constant $C > 0$, such that for all $x_1, x_2 \in \mathcal{X}$, the following inequality holds:*

$$\|\Phi(x_1) - \Phi(x_2)\| \le C\|x_1 - x_2\|.$$

*Here, $\mathcal{X}$ and $\mathcal{Y}$ represent compact Banach spaces embedded in Euclidean spaces $\mathbb{R}^{d_x}$ and $\mathbb{R}^{d_y}$, respectively. The constant $C$ is known as the Lipschitz constant of $\Phi$.*

## A.2 Proof of Theorem 1

**Lemma 1** (Theorem 4.13 (b) in (Berk et al., 2022)). *For $(b, \lambda) \in \mathbb{R}^n \times \mathbb{R}_+$, let $\hat{x}$ be a solution to LASSO problem equation 13 with $I := supp(\hat{x})$. If Assumption 1 holds for $\hat{x}$, then the mapping $S := (b, \lambda) \mapsto \hat{x}$, is locally Lipschitz at the point $(b, \lambda)$ with a Lipschitz constant*

$$L \le \frac{1}{\sigma_{min}(A_I)^2} \left[ \sigma_{max}(A_I) + \|\frac{A_I^\top \bar{r}}{\bar{\lambda}}\| \right] .$$

*Here $\bar{r} := A\hat{x} - b$ represents the residual, and $\sigma_{min}(A_I)$ and $\sigma_{max}(A_I)$ denote the smallest and largest singular values of $A_I$ respectively.*

We can use Lemma 1 and the SVD of submatrices property (Lemma 2) to establish a global regularization result for the regularized inverse map $b \mapsto \hat{x}$.

**Lemma 2.** *For any matrix $X = \begin{bmatrix} A & B \end{bmatrix} \in \mathbb{R}^{m \times n}$, we have*

$$\begin{aligned}
\sigma_{max}(A)^2 &\le \sigma_{max}(X)^2 \le \sigma_{max}(A)^2 + \sigma_{max}(B)^2 \,, \\
\sigma_{min}(A)^2 &\ge \sigma_{min}(X)^2 \ge \sigma_{min}(A)^2 + \sigma_{min}(B)^2 \,.
\end{aligned} \tag{19}$$

*Furthermore, we have*

$$Cond(A) \le Cond(X) \le Cond(A) + Cond(B).$$

*Proof.* Using the triangle inequality, we have

$$\sigma_{\max}(XX^\top) = \|AA^\top + BB^\top\| \le \|AA^\top\| + \|BB^\top\|.$$

Note that

$$\sigma_{\max}(A) = \max_{\|x\|=1 , x_B=0} \|Xx\| \le \max_{\|x\|=1} \|Xx\| = \sigma_{\max}(X).$$

Similarly, we have $\sigma_{\min}(A) \ge \sigma_{\min}(X)$.

$$\begin{aligned}
\sigma_{\min}(XX^\top) &= \min_x x(AA^\top + BB^\top)x^\top \\
&\ge \min_x xAA^\top x^\top + \min_x xBB^\top x^\top \\
&= \sigma_{\min}(AA^\top) + \sigma_{\min}(BB^\top).
\end{aligned} \tag{20}$$

Therefore the second result holds.

For the condition number, we have

$$\begin{aligned}
\mathrm{Cond}(X) &= \frac{\sigma_{\max}(X)}{\sigma_{\min}(X)} \\
&\le \sqrt{\frac{\sigma_{\max}(A)^2 + \sigma_{\max}(B)^2}{\sigma_{\min}(A)^2 + \sigma_{\min}(B)^2}} \\
&\le \sqrt{\frac{\sigma_{\max}(A)^2}{\sigma_{\min}(A)^2 + \sigma_{\min}(B)^2}} + \sqrt{\frac{\sigma_{\max}(B)^2}{\sigma_{\min}(A)^2 + \sigma_{\min}(B)^2}} \\
&\le \mathrm{Cond}(A) + \mathrm{Cond}(B).
\end{aligned} \tag{21}$$

$\square$

*Proof of Theorem 1.* By applying Lemma 1, we can derive the following inequalities:

$$L \leq \frac{\sigma_{\max}(A_I) + 1}{\sigma_{\min}^2(A_I)} \leq \frac{\mathrm{Cond}(A)}{\sigma_{\min}(A)} + \frac{1}{\sigma_{\min}^2(A)} \, .$$

In the first inequality, we utilize the non-degeneracy Assumption 1, and in the last inequality, we employ Lemma 2. $\square$

### A.3   Proof of Theorem 2

*Proof.* We consider two different measurements $m, m' \in \mathcal{M}$ and their regularize inverses $\hat{a} := f_{\mathrm{reg}}^{-1}(m)$ and $\hat{a}' := f_{\mathrm{reg}}^{-1}(m')$. We further denote that

$$\hat{a} = \frac{1}{Z(m)} \int_{\mathcal{A}} a \exp(-\Phi(a; m)) d\mu_0 := \frac{Y(m)}{Z(m)}$$

Then we can decompose the difference of $\hat{a}$ and $\hat{a}'$ as

$$
\begin{aligned}
\|\hat{a} - \hat{a}'\| &\leq \left\| \frac{Y(m)}{Z(m)} - \frac{Y(m')}{Z(m')} \right\| \\
&\leq \frac{1}{Z(m)Z(m')} \left( |Z(m) - Z(m')| \|Y(m)\| + Z(m)\|Y(m) - Y(m')\| \right)
\end{aligned}
\tag{22}
$$

To show $f_{\mathrm{reg}}^{-1}$ is Lipschitz, it suffices to provide a lower bound of $Z$, upper bounds and Lipschitz continuity of $Z$ and $Y$ respectively. We first provide a lower bound of $Z(m)$. Note that

$$Z(m) = \int_{\mathcal{A}} \exp(-\Phi(a; m)) d\mu_0$$

By continuity of $\Phi$ and boundedness of $\mathcal{A}$ and $\mathcal{M}$, we have an upper bound of $\Phi$

$$\sup_{(a,m) \in \mathcal{A} \times \mathcal{M}} \Phi(a; m) = R < \infty \, .$$

This implies that

$$Z(m) \geq \exp(-R) \int_{\mathcal{A}} d\mu_0 = e^{-R} \mu_0(\mathcal{A}) = e^{-R} > 0 \, . \tag{23}$$

Next we provide upper bounds of $Z$ and $Y$ respectively. The upper bound of $Z$ is a consequence of assumption on the lower bound of $\Phi$ and integrable condition on $M_1$.

$$Z(m) \leq \int_{\mathcal{A}} \exp(M_1(\|a\|)) d\mu_0 \leq C \, . \tag{24}$$

where $C = C(M_1) > 0$ is a constant that only depends on $M_1$. We can obtain an upper bound of $Y$ analagously.

$$\|Y(m)\| \leq \int_{\mathcal{A}} \|a\| \exp(M_1(\|a\|)) d\mu_0 \leq C \sup_{a \in \mathcal{A}} \|a\| \, . \tag{25}$$

We next show $Z(y)$ is Lipschitz. By Mean value theorem, we first have

$$
\begin{aligned}
|Z(m) - Z(m')| &\leq \int_{\mathcal{A}} |\exp(-\Phi(a; m)) - \exp(-\Phi(a; m))| d\mu_0 \\
&\leq \int_{\mathcal{A}} \exp(-\lambda \Phi(a; m) - (1-\lambda)\Phi(a; m'))|\Phi(a; m) - \Phi(a; m')| d\mu_0
\end{aligned}
$$

for some $0 < \lambda < 1$. We can then apply assumption (iii) and obtain that

$$\begin{aligned}
|Z(m) - Z(m')| &\leq \int_{\mathcal{A}} \exp(M_1(\|a\|))|\Phi(a; m) - \Phi(a; m')|d\mu_0 \\
&\leq \|m - m'\| \int_{\mathcal{A}} \exp(M_1(\|a\|))M_2(\|a\|)d\mu_0 \\
&\leq C\|m - m'\|
\end{aligned} \tag{26}$$

where $C = C(M_1, M_2) > 0$ is a constant that depends on $M_1$ and $M_2$. Similarly, we can derive the Lipschitz continuity of $Y(m)$.

$$|Y(m) - Y(m')| \leq C \sup_{a \in \mathcal{A}} \|a\| \|m - m'\|, \tag{27}$$

where $C = C(M_1, M_2) > 0$ is the same constant in equation 26. Finally, we plug equation 23, equation 24, equation 25, equation 26 and equation 27 into equation 22.

$$\|\hat{a} - \hat{a}'\| \leq 2e^{-2R} \sup_{a \in \mathcal{A}} \|a\| C(M_1, M_2)\|m - m'\|. \tag{28}$$

$\square$

### A.4   Proof of Theorem 4

*Proof.* Let $r_{\mathcal{M}} = \max_{m \in \mathcal{M}} \|m\|_2$ and define $g : \frac{1}{r_{\mathcal{M}}}\mathcal{M} \to \mathbb{R}^{d_a}$ as

$$g(m) = f_{\text{reg}}^{-1}(m r_{\mathcal{M}}), \quad \forall m \in \frac{1}{r_{\mathcal{M}}}\mathcal{M}.$$

Then $g$ is $L_f r_{\mathcal{M}}$-Lipschitz on $\frac{1}{r_{\mathcal{M}}}\mathcal{M}$. Denote that $g(m) = [g_1(m), \ldots, g_{d_a}(m)]$ then each component $g_i$ is also $L_f r_{\mathcal{M}}$-Lipschitz on $\frac{1}{r_{\mathcal{M}}}\mathcal{M} \subset \mathbb{R}^{d_m}$. Therefore, applying Theorem 3 for each component $g_i$, there exists a ReLU network $\phi_i$ with a width of $d_i = 12d_a \max\{d_m \lfloor p^{1/d_m} \rfloor, p+1\}$ and a depth of $12L + 14 + 2d_m$ such that

$$\sup_{m \in \frac{1}{r_{\mathcal{M}}}\mathcal{M}} |g_i(m) - \phi_i(m)| \leq 19\sqrt{d_m} L_f r_{\mathcal{M}} p^{-2/d_m} L^{-2/d_m}, \quad \forall i = 1, \ldots, d_a.$$

This is equivalent to

$$\sup_{m \in \mathcal{M}} |f_{\text{reg},i}^{-1}(m) - \phi_i(m)| \leq 19\sqrt{d_m} L_f r_{\mathcal{M}} p^{-2/d_m} L^{-2/d_m}, \quad \forall i = 1, \ldots, d_a.$$

This further implies that

$$\sup_{m \in \mathcal{M}} \|f_{\text{reg}}^{-1}(m) - \phi(m)\|_2 \leq 19\sqrt{d_m d_a} L_f r_{\mathcal{M}} p^{-2/d_m} L^{-2/d_m}.$$

Here $\phi(m) = [\phi_1(m), \ldots, \phi_{d_a}(m)]$ represents the neural network that stacks all subnetworks $\phi_i$. Therefore $\phi$ is a neural network with a width of $12d_a \max\{d_m \lfloor p^{1/d_m} \rfloor, p+1\}$ and a depth of $12L + 14 + 2d_m$. $\square$

### A.5   Proof of Theorem 5

*Proof.* The proof can be divided into three distinct parts. Initially, we establish that the testing error is limited by an empirical process. We subsequently demonstrate that the empirical process can be upper bounded by the Rademacher complexity of the neural network function class. Similar ideas have been used in (Györfi et al., 2002). Finally, we provide an estimation for the Rademacher complexity.

**Empirical process bound** Our initial step is to demonstrate that the generalization error is upper-bounded by the empirical process.

$$
\begin{aligned}
\mathcal{R}(\phi_{\theta^*}) - \mathcal{R}(\phi_*) &= \left(\mathcal{R}(\phi_{\theta^*}) - \hat{\mathcal{R}}(\phi_{\theta^*})\right) + \left(\hat{\mathcal{R}}(\phi_{\theta^*}) - \hat{\mathcal{R}}(\phi_*)\right) + \left(\hat{\mathcal{R}}(\phi_*) - \mathcal{R}(\phi_*)\right) \\
&\leq \left(\mathcal{R}(\phi_{\theta^*}) - \hat{\mathcal{R}}(\phi_{\theta^*})\right) + 0 + \left(\hat{\mathcal{R}}(\phi_*) - \mathcal{R}(\phi_*)\right) \\
&\leq 2 \sup_{\phi \in \mathcal{F}_{\mathrm{NN}}} |\hat{\mathcal{R}}(\phi) - \mathcal{R}(\phi)|,
\end{aligned}
\tag{29}
$$

where the nequality is a result of the optimality of $\phi_{\theta^*}$ over the training data, while the second inequality stems from the definition of the supremum. We refer to the quantity $\hat{\mathcal{R}}(\phi) - \mathcal{R}(\phi)$ as the *empirical process*, and it is indexed by $\phi \in \mathcal{F}_{\mathrm{NN}}$.

**Rademacher Complexity bound** Our next objective is to determine an upper bound for $\mathbb{E}\left[\sup_{\phi \in \mathcal{F}_{\mathrm{NN}}} \hat{\mathcal{R}}(\phi) - \mathcal{R}(\phi)\right]$. This bound can be established through the use of the Rademacher complexity, employing techniques such as symmetrization or the "ghost variable" trick. To facilitate this, we define the data set as $\mathcal{S}_{\mathrm{reg}} = \{z_1, \ldots, z_n\}$ and introduce an independent copy denoted as $\mathcal{S}'_{\mathrm{reg}} = \{z'_1, \ldots, z'_n\}$. Note that

$$
\mathcal{R}(\phi) = \mathbb{E}\left[\hat{\mathcal{R}}(\phi, \mathcal{S}_{\mathrm{reg}})\right] = \mathbb{E}\left[\hat{\mathcal{R}}(\phi, \mathcal{S}'_{\mathrm{reg}})\right],
$$

where we use $\hat{\mathcal{R}}(\phi, \mathcal{S}_{\mathrm{reg}})$ to represent the empirical loss of the function $\phi$ on the training data $\mathcal{S}_{\mathrm{reg}}$ and $\hat{\mathcal{R}}(\phi, \mathcal{S}'_{\mathrm{reg}})$ to denote the empirical loss on the training data $\mathcal{S}'_{\mathrm{reg}}$. Therefore

$$
\begin{aligned}
\mathbb{E}\left[\sup_{\phi \in \mathcal{F}_{\mathrm{NN}}} \hat{\mathcal{R}}(\phi) - \mathcal{R}(\phi)\right] &= \mathbb{E}\left[\sup_{\phi \in \mathcal{F}_{\mathrm{NN}}} \hat{\mathcal{R}}(\phi, \mathcal{S}_{\mathrm{reg}}) - \mathbb{E}\left[\hat{\mathcal{R}}(\phi, \mathcal{S}'_{\mathrm{reg}})\right]\right] \\
&= \mathbb{E}\left[\sup_{\phi \in \mathcal{F}_{\mathrm{NN}}} \mathbb{E}\left[\hat{\mathcal{R}}(\phi, \mathcal{S}_{\mathrm{reg}}) - \hat{\mathcal{R}}(\phi, \mathcal{S}'_{\mathrm{reg}}) \mid \mathcal{S}_{\mathrm{reg}}\right]\right]
\end{aligned}
\tag{30}
$$

where the second equality is a result of the independence of $\mathcal{S}'_{\mathrm{reg}}$ from $\mathcal{S}_{\mathrm{reg}}$. Subsequently, we interchange the inner expectation and the supremum, and we can apply the law of total expectation.

$$
\begin{aligned}
&\mathbb{E}\left[\sup_{\phi \in \mathcal{F}_{\mathrm{NN}}} \mathbb{E}\left[\hat{\mathcal{R}}(\phi, \mathcal{S}_{\mathrm{reg}}) - \hat{\mathcal{R}}(\phi, \mathcal{S}'_{\mathrm{reg}}) \mid \mathcal{S}_{\mathrm{reg}}\right]\right] \\
\leq& \mathbb{E}\left[\mathbb{E}\left[\sup_{\phi \in \mathcal{F}_{\mathrm{NN}}} \hat{\mathcal{R}}(\phi, \mathcal{S}_{\mathrm{reg}}) - \hat{\mathcal{R}}(\phi, \mathcal{S}'_{\mathrm{reg}}) \mid \mathcal{S}_{\mathrm{reg}}\right]\right] \\
=& \mathbb{E}\left[\sup_{\phi \in \mathcal{F}_{\mathrm{NN}}} \hat{\mathcal{R}}(\phi, \mathcal{S}_{\mathrm{reg}}) - \hat{\mathcal{R}}(\phi, \mathcal{S}'_{\mathrm{reg}})\right] \\
=& \mathbb{E}\left[\sup_{\phi \in \mathcal{F}_{\mathrm{NN}}} \frac{1}{n}\sum_{i=1}^{n} l(z_i, \phi) - l(z'_i, \phi)\right]
\end{aligned}
\tag{31}
$$

Next we introduce i.i.d. Bernoulli random variables $\sigma_i$. By applying symmetry, we obtain the following:

$$
\begin{aligned}
&\mathbb{E}\left[\sup_{\phi \in \mathcal{F}_{\mathrm{NN}}} \frac{1}{n}\sum_{i=1}^{n} l(z_i, \phi) - l(z'_i, \phi)\right] \\
=& \mathbb{E}\left[\sup_{\phi \in \mathcal{F}_{\mathrm{NN}}} \frac{1}{n}\sum_{i=1}^{n} \sigma_i \left(l(z_i, \phi) - l(z'_i, \phi)\right)\right] \\
=& \mathbb{E}\left[\sup_{\phi \in \mathcal{F}_{\mathrm{NN}}} \frac{1}{n}\sum_{i=1}^{n} \sigma_i l(z_i, \phi) - \sigma_i l(z'_i, \phi)\right] \\
\leq& \mathbb{E}\left[\sup_{\phi \in \mathcal{F}_{\mathrm{NN}}} \frac{1}{n}\sum_{i=1}^{n} \sigma_i l(z_i, \phi)\right] + \mathbb{E}\left[\sup_{\phi \in \mathcal{F}_{\mathrm{NN}}} \frac{1}{n}\sum_{i=1}^{n} -\sigma_i l(z'_i, \phi)\right] \\
=& 2\mathbb{E}\left[\sup_{\phi \in \mathcal{F}_{\mathrm{NN}}} \frac{1}{n}\sum_{i=1}^{n} \sigma_i l(z_i, \phi)\right]
\end{aligned}
\tag{32}
$$

where the first inequality is a property of the supremum, and the final equality arises from the symmetry of Bernoulli random variables.

IIn the preceding derivation, we derive the following upper bound for the expectation of the supremum of the empirical process:

$$\mathbb{E}\left[\sup_{\phi\in\mathcal{F}_{\text{NN}}}\hat{\mathcal{R}}(\phi)-\mathcal{R}(\phi)\right]\leq 2\mathbb{E}\left[\sup_{\phi\in\mathcal{F}_{\text{NN}}}\frac{1}{n}\sum_{i=1}^{n}\sigma_i l(z_i,\phi)\right]$$

We now introduce the Rademacher complexity.

**Definition 4** (Rademacher complexity). *Let's consider a collection of functions denoted as $\mathcal{G}=\{g:\mathcal{D}\to\mathbb{R}\}$. The Rademacher complexity over the data set can be expressed as: $\mathcal{S}_n=\{z_1,\ldots,z_n\}\in\mathcal{D}^n$ is*

$$\mathfrak{R}_n(\mathcal{G})=\mathbb{E}\left[\sup_{g\in\mathcal{G}}\frac{1}{n}\sum_{i=1}^{n}\sigma_i g(z_i)\right]$$

*where the random variables $\sigma_i$ are assumed to be i.i.d. Bernoulli random variables.*

Denote $\mathcal{D}=\mathcal{M}\times\mathcal{A}$ and define $\mathcal{G}=\{l(\cdot,\phi):\mathcal{D}\to\mathbb{R},\quad\phi\in\mathcal{F}_{\text{NN}}\}$, we have

$$\mathbb{E}\left[\sup_{\phi\in\mathcal{F}_{\text{NN}}}\hat{\mathcal{R}}(\phi)-\mathcal{R}(\phi)\right]\leq 2\mathfrak{R}_n(\mathcal{G})\,. \tag{33}$$

**Rademacher Complexity of NN class**  To analyze the Rademacher complexity of the loss function $\mathcal{G}$ indexed by the neural network function class, we treat $\mathcal{G}$ as the image of a transformation applied to $\mathcal{F}_{\text{NN}}$. In the subsequent discussion, we demonstrate that the Rademacher complexity of the loss function class can be bounded by that of the neural network class. The following result can be viewed as an extension of the Talagrand contraction lemma (Ledoux & Talagrand, 1991).

**Lemma 3.** *Consider a function class $H=\{h:\mathcal{D}\to\mathbb{R}\}$ that maps a domain $\mathcal{D}$ to $\mathbb{R}$. Let $l:\mathbb{R}^m\to\mathbb{R}$ be a function that is $L$-Lipschitz in all its coordinates. In other words, for all $x_i\in\mathcal{D}, i=1,\ldots,m$ and $x_j\in\mathcal{D}$ for any $1\leq j\leq n$, we have*

$$|l(x_1,\ldots,x_j,\ldots,x_m)-l(x_1,\ldots,x_j',\ldots,x_m)|\leq L|x_j-x_j'|\,.$$

*Now consider the following function class:*

$$G=\{l(h_1,\ldots,h_m):D\to\mathbb{R}\mid h_i\in H, i=1,\ldots,m\}.$$

*For a fixed set of samples $S_n=\{z_1,\ldots,z_n\}\in\mathcal{D}^n$, we can demonstrate the following inequality:*

$$\mathbb{E}\left[\sup_{l\in G}\frac{1}{n}\sum_{i=1}^{n}\sigma_i l(h_1(x_i),\ldots,h_m(x_i))\right]\leq Lm\mathbb{E}\left[\sup_{h\in H}\frac{1}{n}\sum_{i=1}^{n}\sigma_i h(x_i)\right]\,.$$

We will begin by demonstrating that the square loss function $l(\cdot,\phi)$ is Lipschitz in all its coordinates. Recall that

$$l(z,\phi)=l(z,\phi_1,\ldots,\phi_{d_a})=\|\hat{a}-\phi(m)\|^2=\sum_{i=1}^{d_a}|\hat{a}_i-\phi_i(m)|^2\,,$$

where $\hat{a}_i$ represents the components of $\hat{a}$ and $\phi_i:\mathcal{M}\to\mathbb{R}$ are subnetwork functions. Now consider functions $\phi_i\in\mathcal{F}_{\text{NN}}, i=1,\ldots,d_a$ and $\phi_j'\in\mathcal{F}_{\text{NN}}$, we have

$$
\begin{aligned}
&|l(z,\phi_1,\ldots,\phi_j,\ldots,\phi_{d_a})-l(z,\phi_1,\ldots,\phi_j',\ldots,\phi_{d_a})|\\
&=\left||\hat{a}_j-\phi_j(m)|^2-|\hat{a}_j-\phi_j'(m)|^2\right|\\
&\leq|\phi_j(m)-\phi_j'(m)||2\hat{a}_j-\phi_j(m)-\phi_j'(m)|\\
&\leq 2(r_{\mathcal{A}}+r_{\mathcal{F}_{\text{NN}}^0})|\phi_j(m)-\phi_j'(m)|\,,
\end{aligned}
\tag{34}
$$

where $r_{\mathcal{A}}$ is the radius of the compact space $\mathcal{A}$ and $r_{\mathcal{F}^0_{\mathrm{NN}}}$ is the radius of neural network class $\mathcal{F}_{\mathrm{NN}}$ in the $\infty$-norm. Consequently, the loss function is $2(r_{\mathcal{A}} + r_{\mathcal{F}^0_{\mathrm{NN}}})$-Lipschitz in its coordinates. By applying Lemma 3, we can derive:

$$\mathfrak{R}_n(\mathcal{G}) \leq 2d_a(r_{\mathcal{A}} + r_{\mathcal{F}^0_{\mathrm{NN}}})\mathfrak{R}_n(\mathcal{F}^0_{\mathrm{NN}}), \tag{35}$$

where $\mathcal{F}^0_{\mathrm{NN}}$ is the class of subnetworks. Combining the empirical process bound equation 29, Rademacher bound equation 33, and equation 35, we have

$$\mathbb{E}_{\mathcal{S}_{\mathrm{reg}}}\left[\mathcal{R}(\phi_{\theta^*}) - \hat{\mathcal{R}}(\phi_{\theta^*})\right] \leq 8d_a(r_{\mathcal{A}} + r_{\mathcal{F}^0_{\mathrm{NN}}})\mathfrak{R}_n(\mathcal{F}^0_{\mathrm{NN}}) \tag{36}$$

Next, we will employ the following theorem to bound the Rademacher complexity of $\mathcal{F}^0_{\mathrm{NN}}$.

**Theorem 7** (Theorem 1 of (Golowich et al., 2018)). *Let $H$ be the class of neural network of depth $d$ over the domain $\mathcal{X}$ using weight matrices satisfying $\|W_j\|_{Frob} \leq M_j, 1 \leq j \leq d$ employing the ReLU activation function. Over the data set $\mathcal{S}_n = \{x_1, \ldots, x_n\} \in \mathcal{X}^n$, we can bound the Rademacher complexity as follows:*

$$\mathfrak{R}_n(H) \leq \frac{r_{\mathcal{X}}\sqrt{2\ln 2d}\prod_{i=1}^d M_j}{\sqrt{n}},$$

*where $r_{\mathcal{X}}$ is the radius of compact space $\mathcal{X}$.*

Finally, we have that

$$\mathbb{E}_{\mathcal{S}_{\mathrm{reg}}}\left[\mathcal{R}(\phi_{\theta^*}) - \hat{\mathcal{R}}(\phi_{\theta^*})\right] \leq 8d_a(r_{\mathcal{A}} + r_{\mathcal{F}^0_{\mathrm{NN}}})\frac{r_{\mathcal{M}}\sqrt{2\ln 2L}M_F^L}{\sqrt{n}}. \tag{37}$$

$\square$

### A.5.1  Proof of Lemma 3

*Proof.* We denote the $\sigma$-field generated by $\sigma_1, \ldots, \sigma_i$ as $\mathcal{F}_i$. Then, we have

$$\mathbb{E}\left[\sup_{l\in G}\frac{1}{n}\sum_{i=1}^n \sigma_i l(h_1(x_i), \ldots, h_m(x_i))\right]$$
$$= \frac{1}{n}\mathbb{E}\left[\mathbb{E}_{\sigma_n}\left[\sup_{h_i\in H} u_{n-1}(h_1, \ldots, h_m) + \sigma_n l(h_1(x_n), \ldots, h_m(x_n)) \mid \mathcal{F}_{n-1}\right]\right] \tag{38}$$

where $u_{n-1}(h_1, \ldots, h_m) = \sum_{i=1}^{n-1}\sigma_i l(h_1(x_i), \ldots, h_m(x_i))$. Without loss of generality, let's assume that the supremum is attained at $h_i^+$ when $\sigma_n = 1$ and at $h_i^-$ when $\sigma_n = -1$. Then we have

$$u_{n-1}(h_1^s, \ldots, h_m^s) + sl(h_1^s(x_n), \ldots, h_m^s(x_n))$$
$$= \sup_{h_i\in H} u_{n-1}(h_1, \ldots, h_m) + sl(h_1(x_n), \ldots, h_m(x_n)), \tag{39}$$

for $s = \{-1, 1\}$. We can show that conditioned on $\mathcal{F}_{n-1}$. Then

$$
\begin{aligned}
&\mathbb{E}_{\sigma_n}\left[\sup_{h_i \in H} u_{n-1}(h_1, \ldots, h_m) + \sigma_n l(h_1(x_n), \ldots, h_m(x_n))\right] \\
&= \frac{1}{2}\left[\sup_{h_i \in H} u_{n-1}(h_1, \ldots, h_m) + l(h_1(x_n), \ldots, h_m(x_n))\right] + \\
&\quad \frac{1}{2}\left[\sup_{h_i \in H} u_{n-1}(h_1, \ldots, h_m) - l(h_1(x_n), \ldots, h_m(x_n))\right] \\
&= \frac{1}{2}\left[u_{n-1}(h_1^+, \ldots, h_m^+) + l(h_1^+(x_n), \ldots, h_m^+(x_n))\right] + \\
&\quad \frac{1}{2}\left[u_{n-1}(h_1^-, \ldots, h_m^-) - l(h_1^-(x_n), \ldots, h_m^-(x_n))\right] \\
&= \frac{1}{2}\left[u_{n-1}(h_1^+, \ldots, h_m^+) + u_{n-1}(h_1^-, \ldots, h_m^-)\right] + \\
&\quad \frac{1}{2}\left[l(h_1^+(x_n), \ldots, h_m^+(x_n)) - l(h_1^-(x_n), \ldots, h_m^-(x_n))\right] .
\end{aligned}
\tag{40}
$$

Notice the second term in the last equality can be written as

$$
\begin{aligned}
&\frac{1}{2}\left[l(h_1^+(x_n), \ldots, h_m^+(x_n)) - l(h_1^-(x_n), \ldots, h_m^-(x_n))\right] \\
&\leq \frac{1}{2} L \sum_{i=1}^m s_i(h_i^+(x_n) - h_i^-(x_n)),
\end{aligned}
\tag{41}
$$

where $s_i = \text{sign}(h_i(x_n) - h_i^-(x_n))$. Combining equation 40 and equation 41, we have

$$
\begin{aligned}
&\mathbb{E}_{\sigma_n}\left[\sup_{h_i \in H} u_{n-1}(h_1, \ldots, h_m) + \sigma_n l(h_1(x_n), \ldots, h_m(x_n))\right] \\
&\leq \frac{1}{2}\left[u_{n-1}(h_1^+, \ldots, h_m^+) + L \sum_{i=1}^m s_i h_i^+(x_n)\right] + \\
&\quad \frac{1}{2}\left[u_{n-1}(h_1^-, \ldots, h_m^-) - L \sum_{i=1}^m s_i h_i^-(x_n)\right] \\
&\leq \frac{1}{2} \sup_{h_i \in H}\left[u_{n-1}(h_1, \ldots, h_m) + L \sum_{i=1}^m s_i h_i(x_n)\right] + \\
&\quad \frac{1}{2} \sup_{h_i \in H}\left[u_{n-1}(h_1, \ldots, h_m) - L \sum_{i=1}^m s_i h_i(x_n)\right] \\
&= \mathbb{E}_{\sigma_n}\left[u_{n-1}(h_1, \ldots, h_m) + L \sum_{i=1}^m \sigma_{n,i} h_i(x_n)\right]
\end{aligned}
\tag{42}
$$

Therefore, plugging the above inequality back to equation 38 yields

$$
\begin{aligned}
&\mathbb{E}\left[\sup_{l \in G} \frac{1}{n} \sum_{i=1}^n \sigma_i l(h_1(x_i), \ldots, h_m(x_i))\right] \\
&\leq \frac{1}{n} \mathbb{E}\left[\mathbb{E}_{\sigma_n}\left[u_{n-1}(h_1, \ldots, h_m) + L \sum_{i=1}^m \sigma_{n,i} h_i(x_n) \mid \mathcal{F}_{n-1}\right]\right]
\end{aligned}
\tag{43}
$$

Now applying the same process for all $j = n - 1, n - 2, \ldots, 1$, we can show that

$$
\begin{aligned}
&\mathbb{E}\left[\sup_{l \in G} \frac{1}{n} \sum_{i=1}^{n} \sigma_i l(h_1(x_i), \ldots, h_m(x_i))\right] \\
&\leq \mathbb{E}\left[\sum_{j=1}^{n} \sum_{i=1}^{m} L \frac{1}{n} \sigma_{j,i} h_i(x_j)\right] \\
&= Lm\mathbb{E}\left[\sup_{h \in H} \frac{1}{n} \sum_{i=1}^{n} \sigma_i h(x_i)\right]
\end{aligned}
\tag{44}
$$

$\square$

### A.6 Proof of Theorem 6

*Proof.* Combining Theorem 4 and Theorem 5, we obtain a neural network $\phi_{\theta^*}$ trained from the neural network operator class $\mathcal{F}_{\mathrm{NN}}(d_a, L, pd_a, M)$ with the following properties:

$$
L = \Omega(d_m \tilde{L}), \quad \text{and} \quad p = \Omega(d_m^{\frac{d_m}{d_m-1}} \tilde{p}).
$$

We obtain that

$$
\begin{aligned}
&\mathcal{E}_{\text{learning}} \\
&\leq c_0 d_m d_a L_f^2 r_{\mathcal{M}}^2 p^{-4/d_m} L^{-4/d_m} + c_1 d_a (r_{\mathcal{A}} + M) \frac{r_{\mathcal{M}} \sqrt{\max\{L, d_m\}} M_F^{c_2 \max\{L, d_m\}}}{\sqrt{n}} \\
&\leq c_0 d_m d_a L_f^2 r_{\mathcal{M}}^2 d_m^{\frac{-4(2d_m-1)}{d_m(d_m-1)}} (\tilde{p}\tilde{L})^{-4/d_m} + c_1 d_a (r_{\mathcal{A}} + M) \frac{r_{\mathcal{M}} \sqrt{d_m} \sqrt{\tilde{L}} M_F^{c_2 d_m}}{\sqrt{n}},
\end{aligned}
\tag{45}
$$

where $c_0, c_1, c_2$ are positive absolute constants that may vary from line to line. Without loss of generality, we can select $M = r_{\mathcal{A}} + 1$ to ensure that the approximation theorem remains applicable. Note that $r_{\mathcal{A}} \leq L_f r_{\mathcal{M}}$. We have

$$
\mathcal{E}_{\text{learning}} \leq c d_a d_m^{1/2} L_f r_{\mathcal{M}}^2 \left( L_f d_m^{\frac{d_m^2 - 9d_m + 4}{2d_m(d_m-1)}} (\tilde{p}\tilde{L})^{-4/d_m} + \frac{\sqrt{\tilde{L}} M_F^{c_2 d_m}}{\sqrt{n}} \right).
$$

$\square$

