# OpenReview forum: "Let data talk: data-regularized operator learning theory for inverse problems"
_TMLR — Rejected by TMLR_

### Review · Reviewer_Kyhc · 2026-03-02

**Summary Of Contributions:**

This paper considers solving inverse problems with deep neural networks. Instead of considering the typical supervised paradigm of learning a map between measurements and corresponding ground-truth images, the authors propose to learn a neural network on supervised data consisting of measurements and solutions given by regularized inversion methods. The motivation is that this regularized inversion map is easier to learn due to its smoother structure. The authors analyze two types of inversion maps: those based on variational regularization and Bayesian methods. Theory is presented analyzing generalization error when estimating the map with neural networks and under what conditions the inversion map is Lipschitz.

**Audience:**

No

**Audience Explanation:**

Folks at the intersection of ML and inverse problems would potentially find this work's perspective interesting, but the lack of evidence supporting that the method works well or is useful makes it harder for the work to be of broader interest.

**Claims And Evidence:**

No

**Claims Explanation:**

The submission's contributions are purely theoretical and, while I believe solely theoretical contributions are important, this is an instance in which I think empirical evidence would make the claims much more convincing.

In most instances, papers where the sole contributions are theoretical are typically supporting a founded methodology that has shown to work well empirically, or the theoretical question is interesting in its own right. Here, the authors are proposing a new methodology for inverse problems (training on regularized data) without showing that this approach necessarily shows improvements over, say, purely supervised training or simply regularizing the neural network directly. In particular, many questions about the proposed methodology remain:
- What are the pros and cons of training with data regularized via a variational approach vs bayesian method?
- Can a mixture of explicit + regularized data work well too?
- Does directly regularizing the neural network inversion map yield similar performance to training on regularized data?
- Is robustness empirically observed and does training on regularized data offer benefits with respect to distributional shifts at inversion time?
- Finally, is the neural network's performance essentially bounded by the performance of the regularization method used to obtain regularized data and, if so, what is the main benefit here aside from faster inversion down the line?

Moreover, I would argue that the theoretical results presented here do no carry enough weight on their own as they are somewhat "standard" (e.g., the Lipschitz results use prior work on variational perspectives on LASSO and the generalization error analysis uses standard Rademacher complexity bounds and prior works on generalization error for feedforward ReLU networks when estimating Lipschitz maps). Hence, while the idea has potential, I believe the theoretical contributions alone do not offer enough evidence to support the proposed approach and empirics are needed.

**Requested Changes:**

Crucial changes:
- Experiments analyzing some of the above questions I mentioned would be important in securing my recommendation. I think a more direct comparison between supervising on explicit versus regularized data is the most necessary experiment, along with understanding how the choice of regularization plays a role in properties of the learned neural network.

Helpful changes:
- There are similarities in ideas between this work and those based on unrolling, where the variational regularization method + optimization algorithm inspire the neural network architecture, as opposed to using the actual data itself. One could add a comment about the similarities and differences in the manuscript.

---

### Review · Reviewer_SXAQ · 2026-03-07

**Summary Of Contributions:**

The paper addresses the problem of learning how to invert PDE measurements to PDE parameters (the “inverse problem”). Since the inversion is usually ill-posed, a regularizer needs to be imposed on the neural network. The authors propose to instead train a neural network on paired data where the labels are already regularized solutions. They propose using either Lasso-regularized solutions or solutions coming from a Bayesian posterior. The authors provide some theoretical analysis of the learning and approximation errors of such a network.

The premise of the paper is enticing. The intro suggests that training with regularized data leads to better learned models for solving PDE inverse problems. However, the paper falls short in giving any evidence that such an approach is reasonable, and most of the discussion and theoretical analysis are irrelevant. Here is a list of the main weaknesses:
* There are absolutely no experiments demonstrating the performance of the proposed approach.
* It seems the authors took existing learning error theorems and plugged in $f^{-1}_{reg}$ as the target function, so no theoretical contributions are made.
* The entire framing of the paper is quite strange. For example, the authors could have chosen any other ill-posed inverse problem (besides PDEs) and kept the exact same approach and analysis. The choice of LASSO regularization and Bayesian regularization appear completely arbitrary. It also doesn’t make sense to refer to Bayesian inference itself a regularization technique, since the regularization depends on the chosen Bayesian prior.
* The proposed approach is extremely simple. It simply changes the training data to directly use regularized data rather than enforcing regularization in the training objective. A simple approach is not necessarily a problem, but the problem is that the authors don’t have any experiments to show why this approach is better than enforcing regularization in the training objective.
* The authors do not address why a neural network surrogate is needed if there already exists a way to obtain regularized data (e.g., through LASSO regularization of Bayesian inference).
* The text contains a lot of irrelevant details about learning theory (e.g., in Section 2.1).

**Audience:**

No

**Audience Explanation:**

The paper provides neither a useful approach nor useful information for solving PDE-related inverse problems.

**Broader Impact Concerns:**

Consider discussing the ethical implications of using AI for scientific inverse problems.

**Claims And Evidence:**

No

**Claims Explanation:**

The authors do not provide any empirical evidence showing that training on regularized data is better than training with a regularization term in the training objective.

**Requested Changes:**

The authors should add experiments. These should at least include one PDE inverse problem that researchers care about and at least the baseline of training a neural network with a regularization term in the training objective.

---

### Review · Reviewer_mCKy · 2026-03-10

**Summary Of Contributions:**

This paper introduces the Data-Regularized Operator Learning (DaROL) framework for solving inverse problems. The core idea is a two-stage approach: first, noisy measurement data is "cleaned" by passing it through a classical regularized solver (such as LASSO or a Bayesian conditional mean estimator); second, a neural network is trained in a supervised manner to approximate this regularized inverse mapping. The authors argue that this decoupling of regularization from neural network training offers advantages in flexibility, simplicity, and theoretical tractability compared to methods that embed regularization directly into the network architecture or loss function.

The primary contribution of the work is theoretical. The authors provide:
1. Proofs that, under specific assumptions (non-degeneracy for LASSO; boundedness and integrability for Bayesian inference), the regularized inverse mapping is globally Lipschitz continuous (Theorems 1 and 2).
2. A breakdown of the total learning error into approximation error (controlled by network size) and generalization error (controlled by sample size).
3. Upper bounds for the approximation error (by applying existing universal approximation theorems) and the generalization error (via Rademacher complexity), which are then combined into a comprehensive learning error estimate for the DaROL framework (Theorem 6).

Key Strengths:
-The decomposition of the learning problem into separate regularization and approximation stages is a clean conceptual framework that simplifies theoretical analysis.
-The authors are transparent about the limitations of their analysis, particularly for severely ill-posed problems where the inverse operator lacks Lipschitz continuity.

Key Weaknesses:
-The core idea of "regularize first, then learn" is straightforward and resembles common preprocessing
-While the analysis is sound, several key theorems are direct applications or minor extensions of existing results (e.g., the Lipschitz property of Bayesian estimators, the universal approximation theorem, and the Rademacher complexity bounds). The final combined error bound, while comprehensive, depends on constants and assumptions that render it largely qualitative and difficult to interpret for practical problems.
-The most significant weakness is the complete absence of numerical experiments. Without any demonstration on a concrete inverse problem (e.g., deblurring, tomography), it is impossible to assess whether the theoretical bounds are predictive, whether the method works in practice, or how it compares to relevant baselines like training on raw data or end-to-end regularized approaches.

In summary, the paper presents a theoretically sound analysis of a natural two-stage approach. However, its modest conceptual advance and the lack of any experimental validation make it difficult to assess its true contribution to the field.

**Audience:**

Yes

**Audience Explanation:**

The findings are not of broad, general interest to the entire machine learning community. However, for researchers specifically focused on the theory and application of machine learning to inverse problems, this paper addresses a relevant question and provides a theoretical stepping stone, making it likely that at least some individuals in TMLR's audience would be interested in its findings.

**Broader Impact Concerns:**

This work is purely theoretical and foundational in nature. It does not present any direct applications, experimental results, or deployed systems. As such, I do not foresee any immediate negative ethical issue.

**Claims And Evidence:**

No

**Claims Explanation:**

While the mathematical derivations appear technically correct, the evidence provided to support the paper's central claims is not convincing and, in some critical respects, is entirely absent.

The paper makes several strong claims about the DaROL framework, including that it is "flexible," "efficient," and offers advantages over existing methods. The primary evidence offered to support these claims is theoretical.

Key theorems that form the backbone of the analysis are not novel contributions but rather direct applications of known results. For instance, the Lipschitz property for the Bayesian conditional mean (Theorem 2) is a standard result in the Bayesian inverse problems literature, and the approximation error bound (Theorem 4) is a direct corollary of an existing universal approximation theorem.

The theoretical guarantees, such as the Lipschitz continuity of the regularized map (Theorems 1 and 2), hold only under specific and fairly restrictive assumptions (e.g., non-degeneracy for LASSO, bounded parameter spaces for the Bayesian analysis). The paper does not provide evidence or discussion on how often these assumptions hold in practice, or how the method behaves when they are violated.

The most significant deficiency is the complete lack of empirical validation. The paper advances a practical methodology, yet provides zero experimental evidence to demonstrate that it works. There are no numerical examples, no comparisons to baseline methods, and no analysis of computational trade-offs.

**Requested Changes:**

1. Add Empirical Validation (Critical), where importantly Darol is compared against some baselines. Maybe, Include a brief discussion of the computational cost: how much time does it take to generate the regularized data vs. the training time vs. the test-time inference?

2. Contextualize the approach and how this differ from simple "prepocessing"

3. Weaken or Remove Unsupported Claims

4. Discuss the Tightness and Practical Meaning of the Theoretical Bounds

---

### Decision · Action_Editor_jKjB · 2026-04-16

**Recommendation:** Reject

**Audience:**

Yes

**Audience Explanation:**

The paper considers data-regularized operator learning for solving inverse problems, a topic of interest to TMLR's audience.

**Claims And Evidence:**

No

**Claims Explanation:**

The paper lacks any empirical validation, which is necessary to support its claims of flexibility, efficiency, and practical benefit over existing methods. The theoretical analysis, while technically sound, largely applies known results and does not compensate for the absence of experiments.

All three reviewers pointed to this lack of experimental evidence, and the authors did not respond to the initial reviews.

**Resubmission Of Major Revision:**

The authors may consider submitting a major revision at a later time.